# The relationship between psychological characteristics of patients and their utilization of psychiatric inpatient treatment: A cross-sectional study, using machine learning

Sou Bouy Lo[1,2], Christian G. Huber[1], Andrea Meyer[2], Stefan Weinmann[1,3], Regula Luethi[1], Frieder Dechent[1], Stefan Borgwardt[1,4], Roselind Lieb[2], Undine E. Lang[1], Julian Moeller[1,2]¤*

1 University Psychiatric Clinics (UPK), University of Basel, Basel, Switzerland, 2 Division of Clinical Psychology and Epidemiology, Department of Psychology, University of Basel, Basel, Switzerland, 3 Department of Psychiatry, Psychotherapy and Psychosomatics, Vivantes Hospital Am Urban, Berlin, Germany, 4 Department of Psychiatry and Psychotherapy, University of Lübeck, Lübeck, Germany

¤ Current address: Clinic for Adults, University Psychiatric Clinics (UPK) Basel, Wilhelm Klein-Strasse, Basel
* julian.moeller@upk.ch

**Data Availability Statement:** The ethical approval for this study does not include that the anonymized data may be made freely available in public

## Abstract

High utilizers (HU) are patients with an above-average use of psychiatric inpatient treatment. A precise characterization of this patient group is important when tailoring specific treatment approaches for them. While the current literature reports evidence of sociodemographic, and socio-clinical characteristics of HU, knowledge regarding their psychological characteristics is sparse. This study aimed to investigate the association between patients' psychological characteristics and their utilization of psychiatric inpatient treatment. Patients from the University Psychiatric Clinics (UPK) Basel diagnosed with schizophrenia spectrum or bipolar affective disorders participated in a survey at the end of their inpatient treatment stay. The survey included assessments of psychological characteristics such as quality of life, self-esteem, self-stigma, subjective experience and meaning of psychoses, insight into the disease, and patients' utilization of psychiatric inpatient treatment in the last 30 months. The outcome variables were two indicators of utilization of psychiatric inpatient treatment, viz. "utilization pattern" (defined as HU vs. Non-HU [NHU]) and "length of stay" (number of inpatient treatment days in the last 30 months). Statistical analyses included multiple regression models, the least absolute shrinkage and selection operator (lasso) method, and the random forest model. We included 112 inpatients, of which 50 were classified as HU and 62 as NHU. The low performance of all statistical models used after cross-validation suggests that none of the estimated psychological variables showed predictive accuracy and hence clinical relevance regarding these two outcomes. Results indicate no link between psychological characteristics and inpatient treatment utilization in patients diagnosed with schizophrenia spectrum or bipolar affective disorders. Thus, in this study, the examined psychological variables do not seem to play an important role in patients' use of psychiatric inpatient treatment; this highlights the need for additional research to further examine underlying mechanisms of high utilization of psychiatric inpatient treatment.

repositories. Therefore, the raw data supporting the conclusions of this manuscript were not uploaded for free access, but will be made available by the Center for Psychotic Disorders of the University Psychiatric Clinics Basel (secretariat email: ZPE@upk.ch), without undue reservation, to any qualified researcher.

**Funding:** JM and SW received funding from the Universitäre Psychiatrische Kliniken (UPK) Basel Forschungsfonds, project no. 25036. The funders had no role in study design, data collection and analysis, decision to publish, or preparation of the manuscript.

**Competing interests:** The authors have declared that no competing interests exist.

# Introduction

High utilizers (HU), also known as heavy or frequent users or repeaters, are a group of patients with an above-average use of psychiatric inpatient treatment. Although only approximately 10% to 30% of all inpatients are HU, these clients' use of the healthcare system's resources is well above average [1–3]. However, they seldom receive treatment tailored to their specific characteristics [4].

Prior studies have investigated the high utilization phenomenon to identify characteristics associated with high utilization. Identifying robust characteristics of HU patients is important when tailoring specific treatment approaches for them, to optimize treatment and improve their health and wellbeing [5, 6].

To investigate the high utilization phenomenon, prior studies focused on sociodemographic and socio-clinical patient characteristics and used variables such as length of hospital stay in days, and/or number of hospital admissions within a specific time window as indicators of patients' utilization of psychiatric inpatient treatment. Some of these studies used a cutoff value for these variables to classify patients into HU and Non-High Utilizer (NHU) [7–10]. Studies showed that patients with a high utilization pattern are younger [8, 11–14] than those with low utilization of psychiatric inpatient treatment. Contradictory results were found regarding the sex of HU and NHU patients. Several studies reported a higher prevalence of men among inpatients with high utilization [2, 15, 16], whereas other studies showed a predominance of women [1, 8, 11, 17, 18].

HU have more often been diagnosed with schizophrenia, schizoaffective disorders, and bipolar affective disorders compared with NHU, indicating that the high utilization phenomenon may be specifically relevant to this patient group [3–5, 11–13, 19–22]. Moreover, HU patients showed higher comorbidity rates with personality disorders and substance abuse disorders than NHU patients [2, 3, 6, 9, 10, 23, 24]. However, one study found no differences in diagnoses between HU and NHU inpatients [25].

Homelessness was also associated with high utilization [9, 14, 22, 24]; moreover, a study reported that 68% of the HU patients had problems with living conditions [26]. The lack of an alternative source of psychosocial care such as access to sheltered housing for HU has been reported previously [3, 10, 27]. The majority of all HU patients reported difficulties in their social/interpersonal relationships [14, 18, 23, 24, 26, 28]. Patients with high utilization reported only four contacts in their social networks on an average, which often consisted of family members and medical staff [18, 26]. Furthermore, HU patients showed higher pharmacological non-adherence, more suicidal tendencies [2, 10, 11, 29], higher personal burden [6] and greater severity of psychopathology [13, 25, 28] compared with NHU inpatients. Moreover, poor insight in psychosis was associated with higher utilization of psychiatric inpatient treatment [30, 31].

While the current literature reports evidence on the demographic and socio-clinical characteristics of high utilization described above, the role of psychological characteristics of patients resulting in high utilization remains unclear. For instance, associations between patients' psychological characteristics and their utilization of psychiatric inpatient treatment have rarely been examined. This is a research gap as the inclusion of further patient characteristics in high utilizer research may increase the understanding of the underlying mechanisms of high utilization in the context of psychiatric inpatient treatment, potentially informing clinical interventions in the future.

In this context, quality of life, self-esteem, self-stigma, subjective experience and meaning of psychoses, and insight into the disease are promising candidates to examine associations with utilization of psychiatric inpatient treatment in patients with schizophrenia spectrum or

bipolar affective disorders because they varied depending on the clinical course of these mental disorders in prior studies [32–37]. High severity levels of a mental disorder can generally be considered as an indicator of high utilization [13, 25, 28]. Therefore, we hypothesized an association between the mentioned psychological characteristics and patients' use of psychiatric inpatient treatment, and investigated it through an exploratory study.

### Aim of study

This study aimed to examine associations between psychological characteristics and utilization of psychiatric inpatient treatment in patients with schizophrenia spectrum or bipolar affective disorders from the UPK Basel. Psychological characteristics included measures of quality of life, self-esteem, self-stigma, subjective experience and meaning of psychoses, and insight into the disease.

## Materials and methods

### Ethical approval

The Ethics Committee Northwest and Central Switzerland (EKNZ) (EKNZ BASEC 2017–02203) provided ethical approval for this study. All participants gave written informed consent after receiving information on the study. Protocols of the present study adhere to the principles expressed in the Declaration of Helsinki regarding research with human participants.

### General framework

Data were obtained from the research project "Characteristics of High Utilization in the Psychiatric University Hospital (UPK) Basel," conducted from April 2018 to May 2019 at the UPK Basel in Switzerland. The UPK provides psychiatric treatment for inpatients and outpatients, covering a catchment area of approximately 190,000 people in and around the city of Basel.

### Study population and procedure

We recruited patients from three psychiatric-inpatient wards belonging to the Center for Psychotic Disorders (ZPE) of the Department of Adult Psychiatry UPK Basel. The three wards have an inpatient treatment capacity of 54 beds, and provide diagnosis-specific psychiatric and psychotherapeutic treatment for patients with schizophrenia spectrum disorder (ICD-10: F20-F29) or bipolar affective disorder (ICD-10: F31.1, F31.2). Admission to the ZPE may either be voluntary or involuntary. In the corresponding study period, a trained psychologist responsible for the study's recruitment assessed all patients admitted to one of the three study wards regarding inclusion and exclusion criteria. Inclusion criteria included good knowledge of the German language, an age range between 18 and 65 years, and a diagnosis of schizophrenia spectrum disorder or bipolar affective disorder. Patients who were unable to consent or those with criminal convictions, dementia, psychiatric diseases resulting from organic causes, or a primary diagnosis of substance abuse disorder were excluded. Subsequently, we asked eligible patients at the end of their inpatient treatment stay, i.e., when the psychiatrist in charge of the case announces that discharge management will be initiated for the respective patient, to participate in this study. After receiving information about the project, the participants signed the informed consent and completed the assessment within an average duration of 50 minutes. The psychologist responsible for the study was present during the entire assessment and aided the patients with queries related to the paper-and-pencil questionnaires if necessary. Patients admitted several times to the ZPE during the study period could only participate once in this study.

## Assessments

Assessments included five paper-and-pencil questionnaires on patient psychological characteristics and interviews on utilization of psychiatric inpatient treatment, psychopathology, and sociodemographic information.

**Quality of life.** The World Health Organization Quality of Life (WHOQOL)-BREF measures quality of life through self-report, and includes four domains: *physical health*, *psychological health*, *social relationships*, and *environment*. All items were rated on a five-point Likert scale. Domain scores were converted to a 0–100 spectrum, with higher scores indicating a higher quality of life. The WHOQOL-BREF is a validated and well-established questionnaire. Cronbach's alpha for the German translation indicated good internal consistency within the range of 0.76 to 0.88 [38, 39].

**Self-esteem.** The Rosenberg Self-Esteem Scale (RSE) assesses self-esteem with 10 items on a five-point Likert scale (from 1 = strongly disagree to 5 = strongly agree) [40]. A higher rating indicates a higher level of self-esteem. The total score was calculated for all items. The German adaptation of the self-report questionnaire revealed satisfactory psychometric properties, while the Cronbach's alpha was 0.88 [41].

**Self-stigma.** To detect the reflection of the patients' internalized self-stigma, the Self-Stigma of Mental Illness Scale-Short Form (SSMIS-SF) was used [42]. The short-form of the self-report questionnaire assesses four constructs rated on a nine-point Likert scale: *awareness*, *agreement*, *application* and *hurts-self*. *Awareness* measures patients' consciousness about common stereotypes about people with mental illnesses. *Agreement* assesses the degree of belief that the stereotypes are accurate. *Application* indicates the extent to which patients internalize the stereotypes and apply these to themselves. Consequently, the degree of their decreased self-esteem and self-efficacy is assessed by *hurts-self*. The constructs were formed by summing up the corresponding items. The English original SSMIS-SF provides good internal consistency ranging between 0.69 and 0.87 [43].

**Subjective experience and meaning of psychoses.** Aspects of meaning and experiencing the psychosis were measured using self-report with the Subjective Sense in Psychosis questionnaire (SUSE). The questionnaire was adapted using the framework of a trialogue research project [44, 45]. The shortened version consists of 29 items, through five subscales, measuring the past (*biographical integration*), positive and negative present experiences (*symptoms positive* and *symptoms negative*), and positive and negative expectations of the future (*positive consequences* and *negative consequences*). The answers are rated on a four-point Likert scale. Cronbach's alpha of the English original questionnaire demonstrated good internal consistency of 0.73 to 0.86. The test-retest reliability ranged from 0.66 to 0.84 [46, 47].

**Insight.** Insight into the disease was assessed using Becks Cognitive Insight Scale (BCIS) [48]. This self-report questionnaire assesses patients' cognitive processes and limitations. The sum of nine items captures the subscale *self-reflectiveness* and six items represent *self-certainty*. Answers were rated on a four-point Likert scale (from 0 = strongly not agree to 3 = strongly agree). The English original scale provides good internal consistency, with a Cronbach's alpha level of 0.60 for *self-certainty* and 0.68 for *self-reflectiveness*. The test-retest reliability showed an adequate correlation of 0.60 [49].

**Utilization of psychiatric inpatient treatment.** We assessed retrospective information regarding patients' mental health care utilization during the last 30 months before the current admission using the Client Sociodemographic and Service Receipt Inventory (CSSRI; [50, 51]). To improve the data quality, inpatient treatment information was cross-checked with electronic records of our patients. We used two different indicators to operationalize patients' mental health care utilization pattern. First, we calculated the dichotomous variable

"utilization pattern" by classifying patients into a HU group (HU; ≥180 inpatient treatment days during 30 months and/or ≥3 number of admissions over 18 months) or a NHU group (NHU; <180 inpatient treatment days during the 30 months and/or <3 number of admissions over 18 months). Second, we calculated the continuous variable "length of stay," which indicates the number of inpatient treatment days in the last 30 months for each patient. We used a dichotomous and a continuous indicator of patients' inpatient utilization pattern as preliminary literature revealed no consistent standard in this regard [7–10].

**Psychopathology.**   We assessed the severity of patients' psychopathology using the German form of the Brief Psychiatric Rating Scale (BPRS). The assessment consists of 18 items, which capture different symptoms of psychopathology (e.g., somatic complaints, feelings of guilt, depressed mood, hostility, and grandiosity), and was rated by the attending psychiatrist. The items were evaluated on a seven-point Likert scale (from 1 = not present to 7 = extremely severe). The sum score consists of all items; higher scores indicate a higher degree of psychopathology. The BPRS is one of the most widely used and validated scales in clinical research and provides good psychometric properties. The retest reliability was approximately 0.7, the internal consistency ranged between 0.75 and 0.79, and the inter-rater reliability ranged between 0.56 and 0.87 [52, 53].

**Sociodemographic information.**   We obtained standard sociodemographic information at the beginning of the assessment.

## Statistical analyses

To report descriptive statistics, we chose the first quartile, median, and third quartile for continuous variables as well as absolute numbers and percentages for discrete variables.

In the first set of analysis, we estimated the association between psychological characteristics of the patients and their utilization of psychiatric inpatient treatment through two separate multiple logistic and linear regression models. The dichotomous variable "utilization pattern" (HU vs. NHU) and the continuous variable "length of stay" (number of inpatient treatment days in the last 30 months before the current stay) were outcome variables. Physical health, psychological health, social relationships, and environment (four subscales of the quality of life questionnaire; [38, 39]); self-esteem [40], awareness, agreement, application, and hurts-self (five subscales of the self-stigma questionnaire; [42]); biographical integration, symptoms positive, symptoms negative, positive consequences, and negative consequences (five subscales of the subjective sense in psychosis questionnaire; [44, 45]); as well as self-reflectiveness and self-certainty (subscales of the insight into the disease questionnaire; [48]) were the predictors in both regression models. To reduce risk of confounding, we adjusted these analyses for several a priori defined potential confounders including sex [2, 8, 11, 18], age [2, 11, 13, 14], and psychopathology [13, 25, 28]. Multicollinearity among the predictors and covariates was moderate, with variance inflation factors ranging between 1.3 and 5.2.

Predictive accuracy is the ability of a statistical model to fit well not only with the (training) data it was developed with but also with corresponding new, previously unseen (test) data. Multiple regression models often suffer from overfitting, and therefore have low predictive accuracy [54], especially if the number of predictors is high relative to the number of participants (linear model) or the number of cases (logistic model), as is the case in our study. To avoid model overfit, we ran two additional models that use machine learning algorithms—the least absolute shrinkage and selection operator (lasso) model, and the random forest model. Both models contained the same predictors, outcomes, and adjustments described above. The lasso model uses a variable selection procedure, in which regression coefficients are deliberately shrunk by implying a penalty term to the likelihood function when fitting the model. Consequently, the lasso models are somewhat more biased than those obtained from multiple

regression models but less variable, i.e., they exhibit increased predictive accuracy [55]. Thus, when replicating the current study, predictors whose coefficients have not been shrunk to zero from the lasso are likely to be predictive. The random forest model belongs to the family of decision tree models, which characteristically generate a large number of bootstrapped decision trees, based on random samples of variables. The model has been shown to exhibit high predictive accuracy [56]. Since our two outcome variables were continuous (length of stay) or dichotomous (HU vs. NHU), both the lasso and random forest models were based on a multiple logistic and linear regression model, respectively. To determine the predictive accuracy of the models, we conducted repeated ($n$ = 10) 10-fold cross-validation. We used the root mean square error (RMSE, i.e. the square root of the variance of the residuals) and the R-squared ($R^2$) as the measures of model accuracy for continuous and dichotomous outcomes, respectively.

The outcome–"length of stay"–and the predictors–"application" (self-stigma questionnaire), "hurts-self" (self-stigma questionnaire), and "negative consequences" (subjective sense in psychosis questionnaire)–were all log transformed to approximate normality and homoscedasticity.

All models were run with R, version 3.3.0, including the R package glmnet for the lasso model [57]. A significance threshold of p<0.05 was set for all univariate analyses.

## Results

After screening 533 patients admitted to the ZPE over the recruitment period, we excluded 421 patients as they did not meet inclusion criteria, declined to participate, or because participation was not indicated or feasible. Finally, 112 patients participated and were included in the analyses (see Flowchart Fig 1).

Participants spent a median of 71 days, on an average, in psychiatric inpatient care over the past 30 months before the current admission. We classified 50 of them as HU and 62 as NHU. Demographic and clinical characteristics of participants are summarized in Table 1.

See the correlation matrix of all relevant variables presented in S1 Table.

### Multiple regression models

Coefficients based on the multiple logistic and the multiple linear regression models are depicted in Table 2. The multiple logistic regression model reported a significant association between "utilization pattern" (HU vs. NHU) and psychological health (quality of life), awareness (self-stigma), and hurts self (self-stigma) in our sample. HU patients reported higher psychological health, awareness and hurts-self compared with NHU patients. Model performance for the multiple logistic regression model based on the area under curve ($AUC$) was 0.79 and strongly decreased to 0.59 after cross-validation (an $AUC$ of 0.50 denotes no predictive accuracy at all, i.e., pure guessing).

Results of the multiple linear regression models revealed significant associations between the continuous variable "length of stay" and psychological health (quality of life) and hurts-self (self-stigma); the higher patients' utilization of psychiatric inpatient treatment, the higher were their scores on psychological health and hurts-self. Model performance for the multiple linear regression model based on the root mean square error ($RMSE$) and the variance explained ($R^2$) was 0.88 and 0.22, respectively, and again strongly deteriorated to 1.07 ($RMSE$) and .08 ($R^2$) after cross-validation.

### Lasso models

Coefficients based on the lasso models are depicted in Table 2. Coefficients for the dichotomous outcome "utilization pattern" were comparable with those based on an ordinary logistic

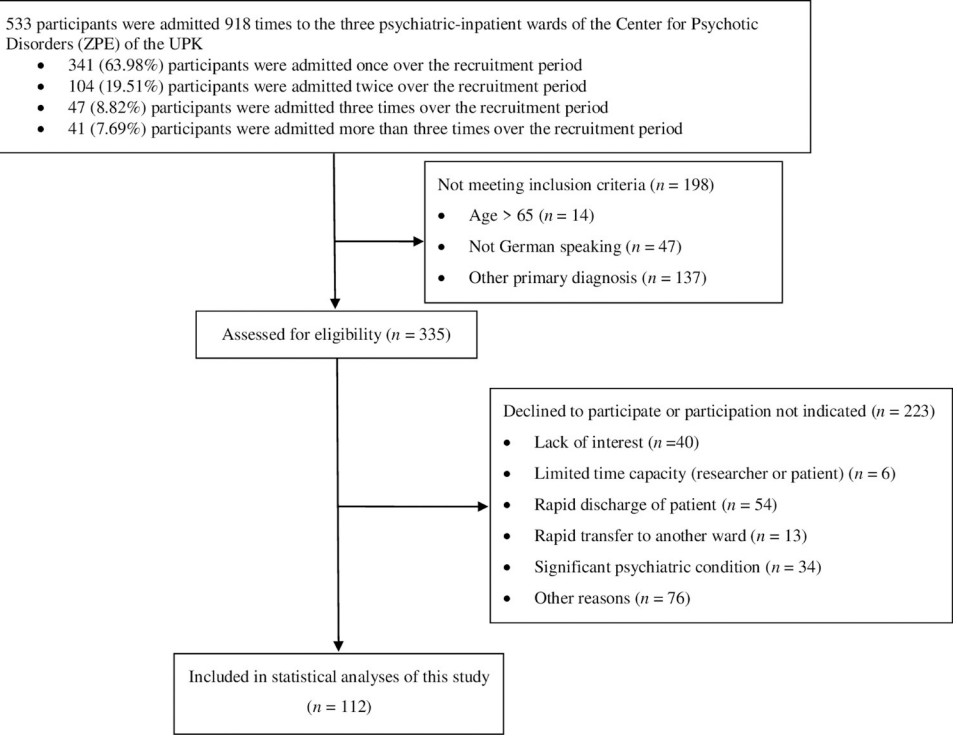

**Fig 1. Flowchart of study participants.**

regression model since the lasso led to the highest predictive accuracy if all predictors were included (and thus none set to 0). However, model performance after cross-validation was low ($AUC$ = 0.58) and not better than that for the cross-validated multiple logistic regression model reported above.

Coefficients based on the lasso model of the outcome "length of stay" were set to 0 for all predictors. They were higher than 0 for only the covariates that were deliberately not shrunk. Thus, none of the predictors based on this model showed predictive accuracy. Model performance after cross-validation ($RMSE$ = 0.99, $R^2$ = 0.10) was slightly better than that of the cross-validated multiple linear regression model. This was, however, due to the three covariates alone since the lasso excluded all predictors.

In sum, the cross validated lasso models suggest that none of the psychological variables tested in this study were able to predict either of the two indicators of patients' utilization of psychiatric inpatient treatment.

## Random forest models

Variable importance values for the two outcomes "utilization pattern" and "length of stay" are shown in Table 2. The most important variables were awareness (self-stigma), and hurts-self (self-stigma) for "utilization pattern" and psychopathology, psychological health (quality of life), awareness (self-stigma), agreement (self-stigma), symptoms positive (subjective experience and meaning of psychoses), and self-reflectiveness (insight into the disease) for length of stay. However, these results must be seen against the background of the low model performance of the random forest model for both outcomes (utilization pattern: AUC = 0.60; length of stay: RMSE = 0.98, R2 = 0.11).

**Table 1. Demographic, clinical, and psychological characteristics of High Utilizer and Non-High Utilizer patients and in the total sample [a].**

| Characteristic | Utilization | | Total |
|---|---|---|---|
| | High Utilizer (*n* = 50) | Non-High Utilizer (*n* = 62) | (*n* = 112) |
| Sex[b] | | | |
| Female | 22.00 (44.00%) | 28.00 (45.16%) | 50.00 (44.64%) |
| Male | 28.00 (56.00%) | 34.00 (54.84%) | 62.00 (55.36%) |
| Age | 42.00 (32.75; 50.25) | 38.00 (28.00; 50.25) | 40.00 (30.00; 50.00) |
| Length of stay in days in the last 30 months | 144.50 (94.50; 224.75) | 35.50 (16.75; 63.25) | 71.50 (29.00; 145.25) |
| Primary diagnosis | | | |
| Bipolar disorder | 12.00 (24.00%) | 14.00 (22.58%) | 26.00 (23.21%) |
| Schizophrenia spectrum disorder | 38.00 (76.00%) | 48.00 (77.42%) | 86.00 (76.79%) |
| Psychopathology | 44.50 (36.00; 50.00) | 40.00 (33.75; 47.00) | 43.00 (35.00; 48.00) |
| Quality of life | | | |
| Physical health | 71.43 (56.25; 78.57) | 67.86 (55.36; 75.00) | 67.86 (57.14; 78.57) |
| Psychological health | 68.75 (53.13; 83.33) | 66.67 (54.17; 75.00) | 66.67 (54.17; 79.17) |
| Social relationships | 66.67 (50.00; 75.00) | 66.67 (50.00; 83.33) | 66.67 (50.00; 75.00) |
| Environment | 68.75 (56.25; 78.13) | 71.88 (59.38; 81.25) | 68.75 (56.25; 81.25) |
| Self-esteem | 37.00 (31.00; 44.25) | 38.00 (34.50; 45.00) | 38.00 (33.00; 45.00) |
| Self-stigma | | | |
| Awareness | 30.00 (24.00; 37.25) | 28.00 (23.00; 33.00) | 29.00 (23.00; 35.00) |
| Agreement | 18.50 (11.50; 23.00) | 17.00 (11.50; 22.00) | 17.00 (12.00; 23.00) |
| Apply | 12.00 (5.00; 18.25) | 11.00 (8.00; 16.50) | 11.00 (7.00; 17.00) |
| Hurts-self | 9.00 (5.00; 20.00) | 7.00 (5.00; 13.00) | 8.00 (5.00; 15.00) |
| Subjective experience and meaning of psychoses | | | |
| Biographical integration | 3.40 (2.60; 3.85) | 3.20 (2.60; 3.80) | 3.40 (2.60; 3.80) |
| Symptoms positive | 3.00 (1.75; 3.43) | 2.75 (2.00; 3.37) | 2.80 (1.80; 3.40) |
| Symptoms negative | 2.79 (1.97; 3.50) | 2.75 (2.06; 3.35) | 2.75 (2.00; 3.38) |
| Positive consequences | 3.40 (2.35; 3.80) | 3.40 (2.90; 3.90) | 3.40 (2.80; 3.80) |
| Negative consequences | 1.83 (1.50; 2.38) | 1.83 (1.50; 2.42) | 1.83 (1.50; 2.37) |
| Insight into the disease | | | |
| Self-reflectiveness | 15.00 (11.00; 18.75) | 15.00 (11.00; 19.00) | 15.00 (11.00; 19.00) |
| Self-certainty | 9.50 (7.25; 12.00) | 9.00 (7.00; 11.00) | 9.00 (7.00; 11.00) |

*Note.* Values based on row data.

[a] If not otherwise specified, median (25th percentile; 75th percentile) is reported

[b] Number (percent) is reported

## Discussion

This study examined the association between specific psychological characteristics and the use of psychiatric inpatient treatment in patients diagnosed with schizophrenia spectrum or bipolar affective disorders from the UPK Basel. While the multiple regression models reported significant associations between specific psychological variables and patients' utilization of psychiatric inpatient treatment, the low model performance of the cross validated multiple regression models combined with the findings of the lasso and the random forest models indicated that none of the psychological variables were actually able to predict patients' utilization of psychiatric inpatient treatment. Hence, our findings do not suggest a link between patients' psychological characteristics estimated in this study and their utilization of psychiatric inpatient treatment. As the values of the respective measures of accuracy (AUC, RMSE, and R2)

**Table 2. Estimated independent effects of psychological characteristics.**

| Variable | Utilization pattern (High Utilizer vs. Non-High-Utilizer) | | | | | | Length of stay (number of inpatient treatment days) | | | | | |
|---|---|---|---|---|---|---|---|---|---|---|---|---|
| | Multiple logistic regression model | | | | Lasso | RF | Multiple linear regression model | | | | Lasso | RF |
| | β | Std. Error | z-value | p | β | VIV | β | Std. Error | t-value | p | β | VIV |
| | (*n* = 112) | | | | | | (*n* = 112) | | | | | |
| Sex | -0.27 | 0.27 | -1.00 | 0.32 | -0.27 | 0.00 | -0.10 | 0.11 | -0.91 | 0.37 | -0.09 | 0.00 |
| Age | 0.26 | 0.27 | 0.99 | 0.32 | 0.26 | 14.96 | 0.01 | 0.11 | 0.11 | 0.91 | 0.03 | 20.49 |
| Psychopathology | 0.58 | 0.27 | 2.19 | 0.03 * | 0.57 | 70.78 | 0.17 | 0.11 | 1.55 | 0.13 | 0.19 | 82.59 |
| Quality of life | | | | | | | | | | | | |
| Physical health | 0.20 | 0.34 | 0.60 | 0.55 | 0.20 | 43.41 | -0.08 | 0.14 | -0.55 | 0.58 | 0.00 | 58.90 |
| Psychological health | 1.26 | 0.52 | 2.41 | 0.02 * | 1.23 | 67.62 | 0.45 | 0.21 | 2.11 | 0.04 * | 0.00 | 84.41 |
| Social relationships | -0.12 | 0.33 | -0.37 | 0.71 | -0.12 | 26.83 | -0.10 | 0.14 | -0.76 | 0.45 | 0.00 | 62.58 |
| Environment | -0.12 | 0.30 | -0.41 | 0.68 | -0.12 | 51.35 | -0.05 | 0.13 | -0.44 | 0.66 | 0.00 | 69.77 |
| Self-esteem | -0.77 | 0.45 | -1.71 | 0.09 | -0.76 | 58.73 | -0.21 | 0.18 | -1.13 | 0.26 | 0.00 | 75.13 |
| Self-stigma | | | | | | | | | | | | |
| Awareness | 0.53 | 0.27 | 1.99 | 0.05 * | 0.52 | 100.00 | -0.03 | 0.11 | -0.26 | 0.80 | 0.00 | 100.00 |
| Agreement | -0.23 | 0.27 | -0.85 | 0.40 | -0.23 | 46.95 | 0.13 | 0.11 | 1.15 | 0.25 | 0.00 | 89.68 |
| Application | -0.62 | 0.38 | -1.62 | 0.11 | -0.60 | 63.57 | -0.25 | 0.15 | -1.73 | 0.09 | 0.00 | 77.84 |
| Hurts-self | 1.05 | 0.44 | 2.40 | 0.02 * | 1.03 | 93.17 | 0.44 | 0.17 | 2.57 | 0.01 * | 0.00 | 77.35 |
| Subjective experience and meaning of psychoses | | | | | | | | | | | | |
| Biographical integration | 0.23 | 0.26 | 0.86 | 0.39 | 0.22 | 36.42 | -0.03 | 0.11 | -0.23 | 0.82 | 0.00 | 58.26 |
| Symptoms positive | 0.61 | 0.39 | 1.58 | 0.11 | 0.59 | 64.55 | 0.14 | 0.15 | 0.96 | 0.34 | 0.00 | 84.20 |
| Symptoms negative | 0.23 | 0.32 | 0.70 | 0.49 | 0.21 | 60.39 | -0.09 | 0.13 | -0.67 | 0.51 | 0.00 | 74.82 |
| Positive consequences | -0.48 | 0.33 | -1.44 | 0.15 | -0.47 | 56.20 | -0.02 | 0.13 | -0.13 | 0.90 | 0.00 | 63.04 |
| Negative consequences | -0.18 | 0.32 | -0.55 | 0.58 | -0.17 | 40.51 | 0.02 | 0.13 | 0.16 | 0.88 | 0.00 | 71.68 |
| Insight into the disease | | | | | | | | | | | | |
| Self-reflectiveness | -0.08 | 0.29 | -0.27 | 0.79 | -0.08 | 60.31 | -0.13 | 0.12 | -1.04 | 0.30 | 0.00 | 81.75 |
| Self-certainty | 0.06 | 0.26 | 0.24 | 0.81 | 0.06 | 46.12 | 0.08 | 0.11 | 0.75 | 0.45 | 0.00 | 68.53 |

*Note*. Values based on transformed data.

Lasso = Least absolute shrinkage and selection operator; RF = Random forest; VIV = Variable importance values; *p* = Significance value. Variable importance values are scaled to lie between 0 and 100.

were insufficient in all the models, this suggests that other patient characteristics–which were not captured in this study–were relevant for utilization of psychiatric inpatient treatment.

To date, prior studies primarily investigated whether sociodemographic and clinical characteristics such as employment, housing or education status (e.g., [9, 22, 23, 26]), social network [18, 26], pharmacological adherence [2, 10, 11, 29], diagnostic information [3, 5, 19, 22], or insight into the disease [31] were related to patients' use of psychiatric inpatient treatment. In contrast, this study examined patients' quality of life, self-esteem, self-stigma, and subjective experience and meaning of psychoses. Thus, this study broadens prior findings by focusing on psychological variables to examine associations between patient characteristics and their utilization of psychiatric inpatient treatment.

In this study, the findings of the cross-validated multiple regression models and the lasso and random forest models indicated that none of the estimated psychological patients' variables showed predictive accuracy regarding patients' utilization of psychiatric inpatient treatment. However, this does not indicate that these psychological patient characteristics are not clinically relevant to patients with schizophrenia spectrum or bipolar affective disorders. In

contrast, there is a growing body of evidence suggesting that self-stigma, self-esteem, and insight into the disease varies among these patients compared to other patient groups. Moreover, these characteristics should be specifically addressed in the psychotherapeutic treatment of patients with psychoses (e.g., [31, 58–62]). In contrast to our findings, prior studies have reported a link between a decreased insight into disease and a higher utilization of psychiatric inpatient treatment in patients with psychosis [30, 31]. However, the findings of this study do not suggest a link between the estimated psychological variables including insight into disease and utilization of psychiatric inpatient treatment. One potential explanation for this observation might be that the examined psychological characteristics of the patients do not vary depending on the use of psychiatric inpatient treatment in patients diagnosed with schizophrenia spectrum or bipolar affective disorders. In this case, there is no evidence that these characteristics should be addressed differently in the psychotherapeutic treatment of HU compared to NHU patients diagnosed with schizophrenia spectrum or bipolar affective disorders. However, this explanation brings up other questions to the fore: whether it is not rather the previously reported factors (e.g., problems in the areas of housing/social network or services that are not tailored to the needs of the patients) that are linked to patients' utilization of psychiatric inpatient treatment, or whether there are other psychological patient characteristics relevant in this context but not examined in this study. Other possible explanations for our observations could be that the methodological aspects in this study ensured that no link between the psychological characteristics of patients and their utilization of psychiatric inpatient treatment was revealed. Potential methodological aspects that can be considered in this context include the comparatively small sample size, potential presence of further confounders that were not included in the analyses, and a potential bias in the estimation of the psychological patient characteristics. It could be, for example, that the self-report questionnaires used in this study did not reliably estimate the patients' psychological characteristics due to their acute psychosis during inpatient treatment. Hence, future studies should be performed in order to either replicate or correct the findings of this study in larger samples. To reduce the risk of a bias, these studies may include further potential confounders such as drug adherence, social support, or substance use in their research design and may investigate patients' psychological characteristics among other less severe psychopathological conditions during outpatient treatment.

Identification of robust psychological variables linked to utilization of psychiatric inpatient treatment in future studies may enhance the understanding of the underlying mechanisms of high utilization and, in turn, foster tailoring of psychotherapeutic interventions to the specific characteristics of HU patients. Describing specific patient groups and tailoring psychiatric and psychotherapeutic interventions to their specific characteristics were discussed as important factors in the implementation of a patient-centered psychiatry with open doors and least restrictive treatment [63–67].

This study has some limitations. First, the sample size is rather small and a substantial proportion of screened patients declined participation. Hence, the findings should be replicated with a larger sample. Second, we measured patients' psychological characteristics in the presence of a researcher; thus, the occurrence of the "interviewer effect" is possible [68]. Accordingly, patients may have trivialized statements, which may have potentially led to underestimating respective variable levels. Third, in this cross-sectional observational study, we entered patients' psychological variables as predictors and "utilization pattern" and "length of stay" as outcome variables into our statistical models. One reason for this was that quality of life, self-esteem, self-stigma, and insight into the disease revealed a relatively high stability over time in prior research [69–73]. These psychological variables may precede and cause patients' behaviors related to their use of psychiatric inpatient treatment. Another reason was that prior observational studies investigating utilization of psychiatric inpatient treatment usually

defined indicators of utilization as outcomes based on the assumption that utilization relates to an outcome concept, which can be influenced by therapeutic interventions (e.g., [5]). In future studies, the use of prospective study designs could clarify the question about whether variables estimating patients' utilization of psychiatric inpatient treatment should be defined as predictors or outcomes. Fourth, another potential limitation is that we cannot rule out the presence of selection bias [74]. As participation was voluntary and inclusion and exclusion criteria were applied, our study sample may possess specific characteristics compared to the respective total patient population, which may limit the generalizability of our findings. Due to these limitations, it is necessary to replicate our findings through further studies before clinical implications for practice can be drawn. Despite these limitations, this study has important strengths. We applied methodologically rigorous "state of the art" statistics to estimate associations between patients' psychological characteristics and their utilization of psychiatric inpatient treatment. These included the lasso model for variable section combined with cross-validation to assess the predictive accuracy of the investigated models based on test data rather than training data. The lasso method considers the family-wise error rate and is, therefore, less likely to cause chance effects [75–77]. Moreover, cross-validation is associated with a higher predictive accuracy, especially when the number of predictors considered is large, relative to the sample size [54]. In this study, using only multiple regression models without cross-validation would have led to several significant associations between psychological variables and utilization, which would likely disappear when replicated. Another strength of the study is the application of two different indicators for utilization of inpatient treatment: the dichotomous classification HU versus NHU, and the continuous variable number of inpatient treatment days during the last 30 months. As prior studies criticized the heterogenous definition of "high utilization," the use of different parameters seemed more appropriate for the statistical analyses of our data [10]. Furthermore, this study examined the high utilization phenomenon in patients diagnosed with schizophrenia spectrum or bipolar affective disorders. High utilization occurs more often in this group of patients compared to patient groups with other primary diagnoses (e.g., [13, 19–22]); thus, we studied the high utilization phenomenon in a particularly relevant population.

In conclusion, our findings do not suggest a link between the estimated psychological characteristics and utilization of psychiatric inpatient treatment in patients with schizophrenia spectrum or bipolar affective disorders. The insufficient understanding of the mechanisms linking specific patients to high utilization, the potentially high personal burden of HU patients and their relatives, as well as the objective to tailor treatment concepts to their specific characteristics highlights the need for greater understanding in this field.

## Supporting information

**S1 Table. Correlation matrix containing bivariate correlations between all relevant variables.**
(DOCX)

## Acknowledgments

The authors want to thank Lukas Imfeld for his help with the extraction of anonymized data and Melanie Nuoffer for her help with the preparation of the tables and literature research.

## Author Contributions

**Conceptualization:** Sou Bouy Lo, Andrea Meyer, Julian Moeller.

**Formal analysis:** Sou Bouy Lo, Julian Moeller.

**Investigation:** Sou Bouy Lo.

**Resources:** Christian G. Huber, Stefan Borgwardt, Undine E. Lang.

**Supervision:** Christian G. Huber, Andrea Meyer, Roselind Lieb.

**Writing – original draft:** Sou Bouy Lo, Andrea Meyer, Julian Moeller.

**Writing – review & editing:** Christian G. Huber, Andrea Meyer, Stefan Weinmann, Regula Luethi, Frieder Dechent, Stefan Borgwardt, Roselind Lieb, Undine E. Lang, Julian Moeller.

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
