## [Decision Letter · Decision Letter 0]

14 Oct 2021

PONE-D-21-06078The relationship between psychological characteristics of patients and their utilization of psychiatric inpatient treatment: A cross-sectional study, using machine learningPLOS ONE

Dear Dr. Moeller,

Thank you for submitting your manuscript to PLOS ONE. After careful consideration, we feel that it has merit but does not fully meet PLOS ONE’s publication criteria as it currently stands. Therefore, we invite you to submit a revised version of the manuscript that addresses the points raised during the review process.

We look forward to receiving your revised manuscript.

Kind regards,

Giuseppe Carrà, PhD

Academic Editor

PLOS ONE

Journal Requirements:

Reviewers' comments:

Reviewer's Responses to Questions

**Comments to the Author**

1. Is the manuscript technically sound, and do the data support the conclusions?

Reviewer #1: Partly

Reviewer #2: Partly

2. Has the statistical analysis been performed appropriately and rigorously? 

Reviewer #1: No

Reviewer #2: No

3. Have the authors made all data underlying the findings in their manuscript fully available?

Reviewer #1: Yes

Reviewer #2: No

4. Is the manuscript presented in an intelligible fashion and written in standard English?

Reviewer #1: Yes

Reviewer #2: Yes

5. Review Comments to the Author

Reviewer #1: This study aimed to explore the psychological characteristics for high utilizers of psychiatric inpatient treatment. The authors enrolled 112 inpatients and classified into 50 as high utilizers and 62 as non-high utilizers. The measured psychological factors included quality of life, self-esteem, self-stigma, subjective experience and meaning of psychosis, and insight. The authors also measured the severity of illness using Brief Psychiatric Rating Scale. The authors used multiple regression model and LASSO model to assess the association between psychological factors and high utilization. The authors found there is no link between psychological characteristics and high utilization.

I think there are several concerns in the methodology.

Firstly, I think the sample size is small. In addition, there are many unmeasured factors, including drug compliance, social support, and substance use. These factors might lead to a null finding due to limited statistic power. Given these limitation, I think the study result is inconclusive.

Secondly, there might be collinearity between these variables. The authors should examine the variance inflation factor. LASSO could be the primary analysis because it might be better than multiple regression model in deal the collinearity problem.

Finally, the authors mentioned the used cross-validation to examine the performance and overfit of model. However, the authors did not mention how they conduct cross-validation.

Reviewer #2: The investigated topic seems both important and practically useful. I found that the authors provided a comprehensive description of the psychological characteristics chosen and that the process for recruiting and interviewing patients seemed sensible. However, I do not feel that training a multiple regression model and LASSO model are sufficient evidence that these psychological characteristics are not predictive of utilization pattern and length of stay. I would have liked to see the authors generate a correlation matrix prior to performing regression to perform the most basic analysis of whether certain features could be predictive.

Both models' performance after cross-validation was poor, nearing random, meaning that based on these models, the authors could not conclude that any of the variables were predictive for the two outcomes. However, there are many predictive methods in machine learning that also offer straightforward methods of extracting feature importance. The authors may be interested in reviewing survey papers such as "Comparing machine learning algorithms for predicting ICU admission and mortality in COVID-19" (Subudhi 2021) "Patient Length of Stay and Mortality Prediction: a survey" (Awad 2017) which describe many methods for prediction of clinical outcomes. For example, in a high performing predictive model of ICU admission, the authors of Subudhi 2021 performed a SHAP analysis on a trained random forest model. SHAP derives from Data Shapley, a method of attributing feature importance to to explain machine learning models. It would be interesting to see the authors of this paper attempt to train a more complex (and hopefully better performing) model and then analyze the psychological features based on a more performant model.

The topic of the paper has clinical significance and the dataset collected by the authors is both novel and useful. However, stronger statistical analysis and the investigation of further machine learning methods would make the paper more compelling towards its conclusion.

6. PLOS authors have the option to publish the peer review history of their article (what does this mean?). If published, this will include your full peer review and any attached files.

Reviewer #1: No

Reviewer #2: **Yes: **Isabel Chien

---

## [Author Response · Author response to Decision Letter 0]

25 Feb 2022

Responses to Reviewer#1:

This study aimed to explore the psychological characteristics for high utilizers of psychiatric inpatient treatment. The authors enrolled 112 inpatients and classified into 50 as high utilizers and 62 as non-high utilizers. The measured psychological factors included quality of life, self-esteem, self-stigma, subjective experience and meaning of psychosis, and insight. The authors also measured the severity of illness using Brief Psychiatric Rating Scale. The authors used multiple regression model and LASSO model to assess the association between psychological factors and high utilization. The authors found there is no link between psychological characteristics and high utilization.

I think there are several concerns in the methodology.

Firstly, I think the sample size is small. In addition, there are many unmeasured factors, including drug compliance, social support, and substance use. These factors might lead to a null finding due to limited statistic power. Given these limitation, I think the study result is inconclusive.

Response: Thank you very much for this helpful comment. For the current study, we recruited subjects from a very vulnerable population – psychotic patients in acute psychiatric care – where it is challenging to conduct studies. Since psychological characteristics have been largely neglected in high-utilizer research and we have applied rigorous statistical analyses, we consider the presented findings as clinically relevant despite the study’s limitations. We hope that the findings of our study will stimulate further research in this field. However, we fully agree with the reviewer that the sample size of this study is rather small with 112 patients included in the analyses. We also agree with the reviewer that we cannot rule out that there may be other potential confounders that we have not been able to consider in the present analyses. Therefore, we have now discussed these potential limitations in relation to our findings in more detail and adapted the revised manuscript accordingly.

Discussion: “Other possible explanations for our observations could be that the methodological aspects in this study ensured that no link between the psychological characteristics of patients and their utilization of psychiatric inpatient treatment was revealed. Potential methodological aspects that can be considered in this context include the comparatively small sample size, potential presence of further confounders that were not included in the analyses, and a potential bias in the estimation of the psychological patient characteristics.”

Discussion: “Hence, future studies should be performed in order to either replicate or correct the findings of this study in larger samples. To reduce the risk of a bias, these studies may include further potential confounders such as drug adherence, social support, or substance use in their research design and …”

Discussion: “Due to these limitations, it is necessary to replicate our findings through further studies before clinical implications for practice can be drawn.”

Secondly, there might be collinearity between these variables. The authors should examine the variance inflation factor.

Response: We thank the reviewer for raising the issue of collinearity and fully agree that the variance inflation factor should be reported. In our study, multicollinearity among the predictors and covariates was only moderate with variance inflation factors ranging between 1.3 and 5.2. A variance inflation factor of >10 is usually used as an indicator of multicollinearity. Since our values are clearly below this threshold, we have refrained from a detailed discussion in the manuscript and have included the sentence below in the revised manuscript.

Statistical analyses: “Multicollinearity among the predictors and covariates was moderate, with variance inflation factors ranging between 1.3 and 5.2.”

LASSO could be the primary analysis because it might be better than multiple regression model in deal the collinearity problem.

Response: We thank the reviewer for his comment, and we fully agree that the machine learning models are the main analyses of the study. However, we had chosen to report the regression models first, as much of the research in the field to date has focused solely on the use of regression models. By using this order of analysis and the resulting "story" we want to point out the importance of rigorous statistical analysis in the context of high utilizer research. We aim to indicate that we might have reached significantly different conclusions in this study if we had focused primarily on the significant associations between utilization pattern and self-stigma of the reported multiple regression models. We would therefore appreciate it very much if the reviewer agrees that we can retain the current order of the statistical analyses in the revised manuscript.

Finally, the authors mentioned the used cross-validation to examine the performance and overfit of model. However, the authors did not mention how they conduct cross-validation.

Response: We thank the reviewer for this comment and have revised the manuscript accordingly.

Statistical analyses: “To determine the predictive accuracy of the models, we conducted repeated (n = 10) 10-fold cross-validation. We used the root mean square error (RMSE, i.e. the square root of the variance of the residuals) and the R-squared (R2) as the measures of model accuracy for continuous and dichotomous outcomes, respectively.”

Responses to Reviewer#2:

The investigated topic seems both important and practically useful. I found that the authors provided a comprehensive description of the psychological characteristics chosen and that the process for recruiting and interviewing patients seemed sensible.

Response: We thank the reviewer for this comment and appreciate his positive feedback on our manuscript regarding the investigated topic, variable selection, and recruitment process.

However, I do not feel that training a multiple regression model and LASSO model are sufficient evidence that these psychological characteristics are not predictive of utilization pattern and length of stay.

Response: We thank the reviewer for this feedback and the specific suggestions, described below in the review, on how we can further improve the rigor of our statistical analyses. To improve the manuscript, we performed additional statistical analyses as suggested by the reviewer and ran a random forest model and included its findings in the revised manuscript. We reported these additional analyses in detail below in our point-by-point response. On the other hand, we discussed in more detail that we cannot completely rule out that methodological aspects were also responsible for the fact that we did not find a link between the psychological patient characteristics and their utilization of psychiatric inpatient treatment. Please see in this regard our point-by-point response to Reviewer 1.

I would have liked to see the authors generate a correlation matrix prior to performing regression to perform the most basic analysis of whether certain features could be predictive.

Response: We agree with the reviewer that a correlation matrix can increase the information content of the manuscript. Accordingly, we included a correlation matrix containing the bivariate correlations between all relevant variables as supporting information in the revised manuscript (supporting table S1, retrievable via hyperlink). We appended the table S1 also below our point-by-point responses.

Results: “See the correlation matrix of all relevant variables presented in supporting information S1 Table .” 

Both models' performance after cross-validation was poor, nearing random, meaning that based on these models, the authors could not conclude that any of the variables were predictive for the two outcomes. However, there are many predictive methods in machine learning that also offer straightforward methods of extracting feature importance. The authors may be interested in reviewing survey papers such as "Comparing machine learning algorithms for predicting ICU admission and mortality in COVID-19" (Subudhi 2021) "Patient Length of Stay and Mortality Prediction: a survey" (Awad 2017) which describe many methods for prediction of clinical outcomes. For example, in a high performing predictive model of ICU admission, the authors of Subudhi 2021 performed a SHAP analysis on a trained random forest model. SHAP derives from Data Shapley, a method of attributing feature importance to explain machine learning models. It would be interesting to see the authors of this paper attempt to train a more complex (and hopefully better performing) model and then analyze the psychological features based on a more performant model. The topic of the paper has clinical significance and the dataset collected by the authors is both novel and useful. However, stronger statistical analysis and the investigation of further machine learning methods would make the paper more compelling towards its conclusion.

Response: We thank the reviewer very much for this comment and for the further literature relating to predictive methods in machine learning. Following the suggestion of the reviewer, we have calculated a second machine learning model, namely a trained random forest model calculating variable importance values for the two outcomes “utilization pattern” and “length of stay”. Due to the low model performance of the random forest model for both outcomes (utilization pattern: AUC=0.60; length of stay: [RMSE=0.98, R2=0.11], the conclusions based on the findings have remained largely the same. We included the trained random forest model as an additional statistical analysis in the manuscript and revised the manuscript accordingly.

Statistical analyses: “To avoid model overfit, we ran two additional models that use machine learning algorithms⎯the least absolute shrinkage and selection operator (lasso) model, and the random forest model. Both models contained the same predictors, outcomes, and adjustments described above.”

Statistical analyses: “The random forest model belongs to the family of decision tree models, which characteristically generate a large number of bootstrapped decision trees, based on random samples of variables. The model has been shown to exhibit high predictive accuracy (Breiman, 2001).”

Breiman L. Random forests. Mach Learn. 2001;45(1):5-32. doi:10.1023/A:1010933404324

Results: “Variable importance values for the two outcomes “utilization pattern” and “length of stay” are shown in Table 2. The most important variables were awareness (self-stigma), and hurts-self (self-stigma) for “utilization pattern” and psychopathology, psychological health (quality of life), awareness (self-stigma), agreement (self-stigma), symptoms positive (subjective experience and meaning of psychoses), and self-reflectiveness (insight into the disease) for length of stay. However, these results must be seen against the background of the low model performance of the random forest model for both outcomes (utilization pattern: AUC=0.60; length of stay: RMSE=0.98, R2=0.11).”

Discussion: “As the values of the respective measures of accuracy (AUC, RMSE, and R2) were insufficient in all the models, this suggests that other patient characteristics – which were not captured in this study – were relevant for utilization of psychiatric inpatient treatment.”

Response: Moreover, we also added several minor changes when we revised the manuscript to include the new findings of the random forest model. For further information, please see our Revised Manuscript with Track Changes. We have included the specific variable importance values of the random forest model in Table 2 of the revised manuscript.

---

## [Decision Letter · Decision Letter 1]

21 Mar 2022

The relationship between psychological characteristics of patients and their utilization of psychiatric inpatient treatment: A cross-sectional study, using machine learning

PONE-D-21-06078R1

Dear Dr. Moeller,

We’re pleased to inform you that your manuscript has been judged scientifically suitable for publication and will be formally accepted for publication once it meets all outstanding technical requirements.

Kind regards,

Giuseppe Carrà, PhD

Academic Editor

PLOS ONE

Reviewers' comments:

Reviewer's Responses to Questions

**Comments to the Author**

1. If the authors have adequately addressed your comments raised in a previous round of review and you feel that this manuscript is now acceptable for publication, you may indicate that here to bypass the “Comments to the Author” section, enter your conflict of interest statement in the “Confidential to Editor” section, and submit your "Accept" recommendation.

Reviewer #1: All comments have been addressed

Reviewer #2: All comments have been addressed

2. Is the manuscript technically sound, and do the data support the conclusions?

Reviewer #1: Yes

Reviewer #2: Yes

3. Has the statistical analysis been performed appropriately and rigorously? 

Reviewer #1: Yes

Reviewer #2: Yes

4. Have the authors made all data underlying the findings in their manuscript fully available?

Reviewer #1: No

Reviewer #2: No

5. Is the manuscript presented in an intelligible fashion and written in standard English?

Reviewer #1: Yes

Reviewer #2: Yes

6. Review Comments to the Author

Reviewer #1: The authors have addressed all my concerns well. I have no further comments on the revised manuscripts.

Reviewer #2: I find that the authors have addressed comments brought up in both reviews appropriately. Again I reiterate that the problem setting seems interesting and significant and although the authors present a negative finding, this work serves as a good initiation into this particular research question. There is potential for more sophisticated machine learning methods to improve the predictive performance and interpretability of the presented models; hopefully these can be explored in follow-up research.

*Note to the authors: This reviewer is a "she," in future it would be good practice to maintain gender neutrality when referencing reviewers.

7. PLOS authors have the option to publish the peer review history of their article (what does this mean?). If published, this will include your full peer review and any attached files.

Reviewer #1: No

Reviewer #2: **Yes: **Isabel Chien

---

## [Editor Report · Acceptance letter]

25 Mar 2022

PONE-D-21-06078R1 

The relationship between psychological characteristics of patients and their utilization of psychiatric inpatient treatment: A cross-sectional study, using machine learning 

Dear Dr. Moeller:

I'm pleased to inform you that your manuscript has been deemed suitable for publication in PLOS ONE. Congratulations! Your manuscript is now with our production department. 

Kind regards, 

on behalf of

Dr. Giuseppe Carrà 

Academic Editor

PLOS ONE